# Morality, self-control, age, type of offence and sentence length as predictors of psychopathy amongst female incarcerated offenders in South Africa

Judite Danielle de Oliveira[1], Jacques Jordaan[1]*, Matthew Cronjé[2]

1 Department of Psychology, University of the Free State, Bloemfontein, Free State, South Africa,
2 Department of Criminology, University of the Free State, Bloemfontein, Free State, South Africa

* JordaanJ1@ufs.ac.za

**Data Availability Statement:** The data can be found at: https://doi.org/10.6084/m9.figshare.24631407

## Abstract

There has been an increase in female incarcerated offenders nationally and internationally. Despite this trend, literature and research on female offenders remain limited compared to their male counterparts. Evidence of the relationship between certain personality disorders and offending behaviour has led numerous countries to prioritise identifying and assessing personality disorders among the offender population. Psychopathic personality traits may contribute to women's risk factors for expressing antisocial behaviours, resulting in their potential future incarceration. Thus, a need exists to understand possible factors that may predict the expression of psychopathic traits in females, which may have notable utility among female offenders. This study aimed to investigate possible predictor variables of psychopathy amongst incarcerated female offenders in South Africa. A quantitative research approach, non-experimental research type, and correlational research design were employed. A convenience sampling technique was used. The sample consisted of 139 ($N =$ 139) female offenders housed in two correctional centres in South Africa who voluntarily participated in this study. Correlation analyses and hierarchical regression analysis procedures were conducted to analyse the results. Results indicated (i) a certain combination of predictor variables that statistically and practically significantly explained both primary and secondary psychopathy and (ii) individual predictor variables (e.g., Impulsivity, Simple Tasks, Risk-Seeking, and Self-Centredness) that explained both primary and secondary psychopathy statistically and practically significantly. This study provides valuable information about the possible predictor variables of psychopathy amongst female offenders within the context of South Africa. However, further research must be conducted to validate these findings and advance our knowledge on this topic.

**Funding:** The author(s) received no specific funding for this work.

**Competing interests:** The authors have declared that no competing interests exist.

## Introduction

Recent years have shown an increase in the number of female incarcerated offenders both nationally and internationally [1–4]. In South Africa, the Department of Correctional Services (DCS) noted an increase from 3 380 in 2012/13 to 4 316 in 2018/19 [5], with the latest count at 3 724 as of March 2022 [6]. In contrast, the population of male offenders decreased exponentially from 150 467 in March 2020 to 139 499 467 by March 2022 [6], possibly due to DCS interventions, including the 2019 presidential remission, the March 2020 National State of Disaster, and the 2020/21 advancement of parole dates [7].

Although focused predominantly on male offenders, research underscores the relationship between personality disorders, particularly psychopathy, and offending behaviour [8–11]. Worldwide, psychopathy rates among incarcerated female offenders range from 9 to 15.5%, contrasting with approximately 31% in males [9, 12–14]. Notably, psychopathic personality traits in women are identified as potential risk factors for antisocial behaviours and future incarceration [9, 15]. These findings purport the need to understand factors influencing psychopathic personality traits in adult females, holding significant clinical relevance for female offenders [16].

Levenson [17] proposes that psychopathy is a pattern of intrinsically antisocial behaviour, driven by judgements regarding one's wishes versus the rights and well-being of others. Levenson et al. [18] distinguish between primary and secondary dimensions of psychopathy, with primary focusing on selfish, uncaring, and manipulative attitudes and secondary emphasizing impulsivity and a self-defeating lifestyle. Previous research links psychopathy to antisocial and offending behaviours [19–21], particularly violent, serious and chronic offences [8, 11, 21]. Female offenders in South Africa commonly engage in economic, aggressive, and violent offences [2, 4, 8, 22] with environmental variables (i.e., poverty) [8, 23], social factors (i.e., social exclusion and lack of policing) [11, 24] and the normalisation of violence [11, 25] often cited as contributing factors. Despite the increasing number of female offenders, research on the connection between psychopathy and offending behaviour, especially recidivism, remains crucial [8, 21, 26]. However, prevailing beliefs about female criminality and lower offending rates compared to males have marginalized the study of female offenders in mainstream psychopathy research [8, 12, 27–29]. Additionally, Douglas et al. [30] and Walters [31] argued that psychopathy may not possess inherent validity in relation to descriptive variables such as age and offender history. DeLisi [19] suggests using psychopathic theories to explore correlates of emerging psychopathic behaviours, allowing researchers to distinguish theories that share similar predictor variables [32] and identify mechanisms connecting psychopathy to offending behaviour. Various variables, including a history of victimisation, aggressive behaviours, personality disorders, substance use, and criminal thinking styles are proposed as potential mediators in explaining the relationship between psychopathy and offending, ultimately aiding in the prediction of psychopathy [8, 10, 11, 33–36]. Moreover, variables such as morality, self-control, age, type of offence and sentence length can serve as possible predictors of psychopathy [8, 12, 21, 37, 38].

Morality, specifically moral metacognition, involves an individual's awareness, monitoring, reflection, and regulation of their thinking within a moral reasoning framework [39, 40]. Moral metacognition is crucial for effective ethical decision-making, contributing to appropriate ethical behavioural outcomes [39–42]. Malatesti et al. [43] argue that psychopaths, characterized by a lack of empathy and resistance to other's emotional influence, tend to reject societal norms [43, 44]. Research further associates psychopathic behaviour with diminished neural reactivity when judging moral transgressions [45–48], resulting in persistent aggressive

behaviours [21]. The observed lack of aversive response to moral transgressions reinforces the link between psychopathy and morality [21].

Self-control, a key construct in the general theory of offending behaviour, is defined as the susceptibility to momentary temptations [49, 50]. Gottfredson and Hirschi [51] posit that an individual's self-control influences offending behaviour when presented with opportunities shaped by structural or situational circumstances [49]. Van Gelder and De Vries [52] suggested that individuals are more likely to offend when perceived benefits outweigh perceived losses, particularly if low self-control leads to prioritizing immediate gratification over long-term consequences [51, 53]. Gottfredson and Hirschi [51] distinguish self-control from a lasting propensity to offend, emphasizing a persistent inclination to ignore long-term consequences, resulting in impulsivity, feebleness, recklessness, and indifference to others' needs [49]. Armstrong et al. [53] note that psychopathy often manifests as callous egocentricity, stemming from a lack of affect rooted in physiological and neuropsychological emotional under-arousal [54, 55]. The connection between psychopathy, self-control, and offending behaviour suggests that exploring this relationship may establish the analytical validity of self-control and psychopathy as distinct aspects of offender propensity [37, 53, 56, 57]. DeLisi and Vaughn [58] argue that the inclination for offending behaviour is predominantly shaped by a psychopathic personality, low self-control, and poor morality. According to Van Gelder and De Vries [59], individuals with a proactive willingness to exploit others for personal gain, displaying self-enhancing and amoral behaviours, are prone to rule violations due to lower moral standards. These behaviours predict psychopathy, self-centeredness [60, 61], and engagement in offending behaviour [52]. Thus, psychopathy, poor morality and low self-control may collectively contribute toward the predisposition to offending behaviours.

Various demographic, genetic and environmental risk factors may contribute to psychopathy [62, 63]. Studies exploring age, type of offense, and sentence length in association with psychopathy reveal limited research on age differences among individuals with aversive personality traits, often focusing on limited age ranges [64–67]. The age of female offenders typically ranges between 19 to 55 years, with an average of 33.5 years [12]. Severity and frequency of psychopathic offending decrease with age among females [8, 21], aligning with the maturity principle proposed by Roberts et al. [68]. Improvements in age-associated adjustment and emotional regulation lead to increased agreeableness, conscientiousness, and emotional stability [66, 69]. Psychopathic offenders habitually commit more non-violent and violent offences from late adolescence to their late forties [70], with a considerably higher number of sentences before the age of 17 [71, 72]. Psychopathic individuals are also more likely to take on a wider range and higher rate of violent behaviours compared to non-psychopathic offenders [8, 26, 73]. Hicks et al. [71] noted differences between primary and secondary psychopathy, with affective and interpersonal psychopathic traits associated with more non-violent offences and behavioural traits linked to more violent offences [71]. Given that sentence length is typically an indication of the seriousness of an offence(s), longer sentences tend to be imposed on individuals with multiple convictions and those who have committed more serious/violent crimes [74, 75].

With the high rate of females currently incarcerated over and above the predicted proliferation of offenders in the coming years [5, 6], it is vital to investigate factors that may predict offending behaviour amongst females in order to support these individuals, regardless of the contraventions that placed them there [76]. Therefore, this research aimed to investigate the possible individual predictor (independent) variables (i.e., morality, self-control, age, type of offence, sentence length) or combination(s) of predictor variables that explain a significant percentage of the variance in psychopathy amongst female incarcerated offenders.

## Methodology

### Research design and approach

A quantitative research approach and a non-experimental type of research were used. The main goal was to describe and measure the direction, strength, and practical significance of correlations between two or more quantitative variables [77, 78], and therefore, a correlational research design [79] was used.

### Research sample

A non-probability sampling technique known as convenience sampling [79, 80] was utilised in this study as this sampling method was determined to pose the least amount of risk to participants and correctional officials. This research was conducted in two South African correctional centres, namely the Bizzah Makhate Correctional Centre (BMCC) in Kroonstad, Free State, and the Johannesburg Correctional Centre (JCC) in Johannesburg, Gauteng. In total, 139 incarcerated female offenders ($N$ = 139) between the ages of 20 and 68 voluntarily participated in the research. Participants of all ages, ethnicities, linguistic groups, religious or spiritual orientations, types of offences, and sentence lengths were included to form part of the sample. Offenders 18 years of age or younger were excluded from the sample.

**Participants.** The frequencies for the research sample, as illustrated in Table 1, were calculated for the participants' type of offence, sentence length, offender type, gang involvement, age, ethnicity, previous psychiatric disorder, and substance abuse.

Based on the demographic information, most participants ($n$ = 105; 75.5%) were sentenced for violent offences such as murder and sexual offences. In comparison, 24.5% ($n$ = 34) were incarcerated for non-violent offences such as fraud, forgery, and theft. Furthermore, the

**Table 1. Frequency distribution of participants according to demographic variable.**

| Biographical Variable | *n* | *%* |
|---|---|---|
| *Type of Offence* | | |
| Non-violent | 34 | 24.5 |
| Violent | 105 | 75.5 |
| *Offender Type* | | |
| First-time offender | 127 | 91.4 |
| Repeat offender | 12 | 8.6 |
| *Gang Involvement* | | |
| No | 130 | 93.5 |
| Yes | 9 | 6.5 |
| *Ethnicity* | | |
| Black (African) | 108 | 77.7 |
| Coloured (mixed race) | 6 | 4.3 |
| Indian | 3 | 2.2 |
| White (Caucasian) | 21 | 15.1 |
| Other | 1 | .7 |
| *Previous Psychiatric Disorder* | | |
| No | 116 | 83.5 |
| Yes | 23 | 16.5 |
| *Substance Abuse* | | |
| No | 101 | 72.7 |
| Yes | 38 | 27.3 |

majority of the participants were first-time offenders ($n$ = 127; 91.4%). Almost all participants ($n$ = 130; 93.5%) indicated that they do not belong to a gang in the correctional environment. Moreover, most participants were between 30 and 39 years old ($n$ = 58; 41.73%). More than three quarters of the participants were Black (African) offenders ($n$ = 108; 77.7%), while the remaining offenders were White (Caucasian) ($n$ = 21; 15.1%), Coloured (individuals of a mixed race) ($n$ = 6; 4.3%), and Indian ($n$ = 3; 2.2%). Regarding previous psychiatric disorders and substance abuse, the majority of the participants were not previously diagnosed with psychiatric disorders ($n$ = 116; 83.5%) and did not abuse any substances ($n$ = 101; 72.7%). The average age of the participants was 35.22 years ($SD$ = 9.602), and their average sentence length was 14.84 years ($SD$ = 33.205).

## Data collection procedures

Approval to conduct this research was first obtained from the General and Human Research Ethics Committee at the University of the Free State with ethical clearance number: UFS-HSD2019/0369/1308. Further approval to conduct this study was obtained from the Department of Correctional Services, South Africa. The research was explained in its entirety to several groups of incarcerated offenders who had been collected and assembled in predetermined venues. These offenders were invited to participate in the study. A total of 89 ($n$ = 89) offenders from BMCC and fifty ($n$ = 50) offenders from JCC chose to participate in the study. All participants were well-versed on their rights as research participants, both verbally and in writing, and that their participation in the research would not influence their sentence and parole outcome in any way.

Participants were also informed that they would not receive any benefits or privileges for their participation in the study. In summary, data collection occurred following the researchers' request that participants sign the informed consent sheet if they wanted to participate in the study. Data collection commenced during 19 November 2021 and was completed during 17 March 2022. The researchers visited the correctional centres on different occasions to recruit participants.

## Measurement scales

Each participant received a single booklet comprised of four distinct questionnaires. The questionnaires took one to one and a half hours to complete depending on the literacy level of the offenders. The researchers were always present to assist the participants in the completion of the questionnaires and created an environment conducive to the offenders feeling free to ask questions and request assistance. The researchers also ensured that the participants completed the questionnaires in a quiet and private space to reduce distractions and improve comprehension. The questionnaires were generated on EvaSys, an automated survey software programme that enabled the researchers to craft the questionnaire through a simple and efficient design to increase participation responses [81]. The instruments used to gather the necessary data included: (i) a self-compiled demographic questionnaire, (ii) the *Levenson Self-Report Psychopathy Scale* (LSRP), (iii) the *Moral Metacognition Scale* (MMS), and (iv) the *Self-Control Scale-Modified* (SCSM).

Each participant completed a self-compiled demographic questionnaire. The questionnaire included items relating to offender demographics such as age, type of offence, sentence length, prior incarceration experience, gang affiliation, diagnosis of a psychiatric disorder, and substance abuse.

The *Levenson Self-Report Psychopathy Scale* (LSRP) [18] was utilised to measure the psychopathy of incarcerated female offenders. The LSRP consists of 26 items, each endorsed according

to a four-point Likert-type scale with response options ranging from 1 ("*strongly disagree*") to 4 ("*strongly agree*") [18]. Some items are reverse-scored to control for rater response sets [82, 83]. The LSRP focuses on two distinct dimensions of psychopathy, namely primary psychopathy and secondary psychopathy [18]. According to offender-based studies, adequate to good internal consistencies for the LSRP primary dimension and secondary dimension, ranging from .83-.84 and .69-.77, respectively, have been reported [16, 84–89]. Acceptable internal consistencies were obtained in this study, namely .73 for primary psychopathy and .69 for secondary psychopathy. This is in line with previous studies. Using the two-dimensional model, higher scale scores suggest greater self-reported propensities for primary and/or secondary traits related to the construct of psychopathy [90]. Therefore, lower scale scores suggest diminished self-reported predispositions for primary and/or secondary psychopathic traits [90].

The *Moral Metacognition Scale* (MMS) [39] was utilised to measure the extent to which offenders monitor and regulate their thinking concerning ethical decision-making. The MMS has 20 items and four subscales, namely (i) Regulation of Cognition, (ii) Knowledge of Cognition (Declarative), (iii) Knowledge of Cognition (Procedural), and (iv) Knowledge of Cognition (Conditional). The items of the MMS are scaled on a six-point Likert-type scale, with response options ranging from 1 ("*very strongly disagree*") to 6 ("*very strongly agree*") [39]. The MMS produced adequate to exceptional internal consistencies for all subscales in previous studies, ranging from .78-.83, .82-.89, .79-.84, and .83-.90 for the Regulation of Cognition, Knowledge of Cognition (Declarative), Knowledge of Cognition (Procedural) and Knowledge of Cognition (Conditional) subscales, respectively [39, 91]. Internal consistencies ranging from .69 for Regulation of Cognition, .75 for Declarative Knowledge of Cognition, .54 for Procedural Knowledge of Cognition, and .60 for Conditional Knowledge of Cognition were obtained in this study. These are lower compared to the findings of previous studies. Higher scores on the MMS suggest that offenders self-report that they employ high levels of cognitive expenditure as they monitor, reflect on, and regulate their thinking during the ethical decision-making process [39]. Therefore, lower scores on the MMS suggest less overall self-reported cognitive expenditure of the offenders in their thought processes when making ethical decisions [39].

The *Self-Control Scale-Modified* (SCSM) [50] was utilised to measure several aspects of offenders' self-control. The SCSM measures 24 items of self-control, divided into six subscales assessing four items each, namely (i) Impulsivity, (ii) Simple Tasks, (iii) Risk-Seeking, (iv) Physical Activities, (v) Self-centredness, and (vi) Temper [51]. Each item on the SCSM is rated on a seven-point Likert-type scale with response options ranging from 0 ("*strongly disagree*") to 6 ("*strongly agree*") [50]. For the purpose of this study, the element of physical activity was excluded from the questionnaire since this factor was not particularly relevant to the overall aim of the study. According to a study using a male-specific offender and undergraduate sample, the internal consistencies of the six subscales equalled .67 (Impulsivity), .73 (Simple Tasks), .74 (Risk-Seeking), .76 (Physical Activity), .73 (Self-centredness) and .79 (Temper), respectively [92]. The internal reliability of each factor of the SCSM in this study was calculated as .59 for Impulsivity, .74 for Simple Tasks, .61 for Risk Seeking, .77 for Self-Centredness, and .76 for Temper, which compares with the findings from previous studies. Using the six-factor model, higher subscale scores on the SCSM indicate low self-control among incarcerated offenders [50]. Low self-control entails certain characteristics, including risk-taking, impulsivity, lacking empathy, preferring simple and easy tasks, and preferring physical tasks [50]. According to Hirschi and Gottfredson [93], these are the same characteristics of criminality. Therefore, low self-control represents the propensity to engage in offending behaviour.

## Data analyses

All data collected from the participants were analysed using the Statistical Package for the Social Sciences (SPSS, Version 28) [94]. Descriptive statistics for the LSRP, MMS and SCSM scales were calculated, including the biographical characteristics of the sample of incarcerated female offenders. Cronbach alpha coefficients were calculated in order to establish the internal consistencies of the various scales [95, 96] followed by Pearson product-moment correlation coefficients that were calculated to determine the strength, direction and significance of the correlations between variables. In order to determine which individual predictor variable(s) (e.g., morality, self-control, age, type of offence and sentence length) or combination(s) of predictor variables explain the highest percentage of variance in psychopathy amongst female incarcerated offenders, hierarchical multiple regression analyses were conducted. Furthermore, the variance in the criterion variable was examined by evaluating the effect size ($f^2$) of the contribution made by an individual predictor or combination of predictors.

## Ethical considerations

As this study endeavoured to obtain sensitive information from a population of female incarcerated offenders, who are considered to be a vulnerable population [2, 97], ethical considerations were paramount throughout the research process. Written informed consent [78] was obtained from participants at the beginning of the study to ensure voluntary participation. Additionally, the researchers maintained voluntary participation by consistently communicating to participants that they could decline participation or withdraw from the study at any point, with no adverse consequences throughout the entire research process. As there were also no direct benefits offered to participants due to their participation in the study, individuals who opted not to participate were not disadvantaged in any way. The study had the potential to cause emotional and/or psychological distress to participants. Consequently, the researchers took measures to ensure the availability of a psychologist and/or a social worker who could provide debriefing or counselling services to participants experiencing emotional distress as a result of the study. Importantly, participants' identifying information was not required to complete the questionnaires. Moreover, a coding system was utilised to ensure the anonymity of participants. All data obtained for the study were stored securely, which further helped protect participants' anonymity and/or confidentiality.

## Results

### Means, standard deviations, skewness, kurtosis and internal consistencies of the various measurement scales

The means, standard deviations, skewness, kurtosis as well as internal consistencies of the various subscales of the measuring instruments are illustrated in Table 2 for the total group of participants. Cronbach's alpha coefficient (α) was calculated as an indication of the internal consistencies of the subscales.

Table 2 illustrates that the internal consistencies for the LSRP, MMS and SCSM subscales range from .54 to .77. The majority of these scales, therefore, displayed acceptable levels of internal consistency [98] and were included in the subsequent analyses. However, the Knowledge of Cognition (Procedural) subscale was excluded from further statistical analysis in this study, as it had an unacceptable level of internal consistency (.54). As part of the descriptive statistics in this table, it was investigated whether the data were normally distributed by calculating the skewness and kurtosis values of the different subscales. According to Kahane [99],

**Table 2. Descriptive statistics and reliability coefficients for the LSRP subscales, MMS subscales, and SCSM subscales.**

| Measures | N | M | SD | α | Skewness | Kurtosis |
|---|---|---|---|---|---|---|
| *LSRP* | | | | | | |
| Primary Psychopathy (PP) | 139 | 2.1763 | .49128 | .73 | -.392 | -.653 |
| Secondary Psychopathy (SP) | 139 | 2.3755 | .59464 | .69 | -.155 | -.640 |
| *MMS* | | | | | | |
| Regulation of Cognition (ROC) | 139 | 4.2386 | .85645 | .69 | -.977 | 1.529 |
| Knowledge of Cognition (D) | 139 | 4.3201 | .97597 | .75 | -1.076 | 1.447 |
| Knowledge of Cognition (P) | 139 | 4.2086 | .88751 | .54 | -1.163 | 2.611 |
| Knowledge of Cognition (C) | 139 | 4.2266 | .96068 | .60 | -.714 | .392 |
| *SCSM* | | | | | | |
| Impulsivity (I) | 139 | 3.8597 | 1.31019 | .60 | -.236 | -.619 |
| Simple Tasks (ST) | 139 | 3.6655 | 1.49987 | .74 | -.034 | -1.138 |
| Risk-Seeking (RS) | 139 | 3.3435 | 1.37006 | .61 | .186 | -.862 |
| Self-Centredness (SC) | 139 | 3.2194 | 1.51844 | .77 | .386 | -.862 |
| Temper (T) | 139 | 3.5863 | 1.58563 | .76 | -.001 | -1.185 |

the cut-off point for skewness is $> |2|$ and kurtosis $> |4|$. Table 2 shows that the scores on all the subscales are within these cut-off points and do not deviate substantially from normal.

## Correlations between variables

The Pearson product-moment correlation coefficients were calculated for the independent (predictor) variables, namely Morality, Self-Control, Age, Type of Offence (non-violent versus violent) and Sentence Length, as well as the dependent (outcome) variable, namely Psychopathy, prior to conducting the regression analyses. All the assumptions of correlational analyses were met. The focus of the discussion will be conserved to correlations of statistical and practical significance. Both the 1% and 5% levels of significance were considered. According to Steyn [100], for correlations, an effect size of .10 is considered small, .30 is medium, and .50 is large. However, only findings with effect sizes of .30 or greater will be discussed. The correlation coefficients are illustrated in Table 3.

Table 3 demonstrates significant positive correlations between primary psychopathy, secondary psychopathy, and various factors. Both primary and secondary psychopathy exhibit statistically significant positive correlations with impulsivity, simple tasks preference, risk-seeking behaviour, self-centeredness, and temper. Additionally, both primary and secondary psychopathy show statistically significant negative correlations with age.

For instance, higher scores on primary psychopathy and secondary psychopathy are associated with elevated impulsivity, that might suggest that incarcerated female offenders with these psychopathy traits tend to display higher impulsivity. Similarly, a preference for simple tasks over complicated activities is linked to higher scores on both primary and secondary psychopathy. Moreover, risk-seeking behaviour seems to be more prevalent among incarcerated female offenders scoring higher on primary and secondary psychopathy, indicating a possible tendency for engagement in risky activities. Elevated self-centeredness is observed in individuals with higher scores on both primary and secondary psychopathy, implying a lack of sympathy among female incarcerated offenders with these traits. Additionally, individuals scoring high on primary and secondary psychopathy might be more likely to have short tempers. Finally, both primary and secondary psychopathy exhibit negative correlations with age, suggesting that younger incarcerated female offenders tend to score higher on these psychopathy dimensions.

**Table 3. Correlations between the LRSP subscales and age, type of offence, sentence length, MMS subscales, and SCSM subscales (N = 139).**

|         | 1 | 2       | 3       | 4        | 5       | 6       | 7       | 8       | 9       | 10      | 11      | 12     | 13      |
|---------|---|---------|---------|----------|---------|---------|---------|---------|---------|---------|---------|--------|---------|
| **1. PP**  | - | .532**  | -.210*  | -.288**  | -.125   | .528**  | .537**  | .462**  | .617**  | .386**  | -.368** | -.011  | -.292** |
| **2. SP**  |   | -       | -.146   | -.265**  | -.122   | .631**  | .594**  | .537**  | .536**  | .649**  | -.320** | .047   | -.248** |
| **3. ROC** |   |         | -       | .782**   | .598**  | -.064   | -.055   | .066    | -.031   | -.042   | -.028   | .009   | .027    |
| **4. D**   |   |         |         | -        | .615**  | -.142   | -.156   | -.036   | -.113   | -.118   | .049    | .027   | .083    |
| **5. C**   |   |         |         |          | -       | -.022   | -.037   | .048    | .023    | -.083   | -.023   | -.110  | -.061   |
| **6. I**   |   |         |         |          |         | -       | .650**  | .471**  | .544**  | .580**  | -.283** | .089   | -.179*  |
| **7. ST**  |   |         |         |          |         |         | -       | .322**  | .514**  | .479**  | -.212*  | -.013  | -.214*  |
| **8. RS**  |   |         |         |          |         |         |         | -       | .611**  | .617**  | -.337** | .030   | -.122   |
| **9. SC**  |   |         |         |          |         |         |         |         | -       | .544**  | -.308** | .052   | -.117   |
| **10. T**  |   |         |         |          |         |         |         |         |         | -       | -.354** | .187*  | -.072   |
| **11. A**  |   |         |         |          |         |         |         |         |         |         | -       | -.188* | .261**  |
| **12. TO** |   |         |         |          |         |         |         |         |         |         |         | -      | .047    |
| **13. SL** |   |         |         |          |         |         |         |         |         |         |         |        | -       |

Key: PP = Primary Psychopathy, SP = Secondary Psychopathy, ROC = Regulation of Cognition, D = Knowledge of Cognition (Declarative), C = Knowledge of Cognition (Conditional), I = Impulsivity, ST = Simple Tasks, RS = Risk-Seeking, SC = Self-Centredness, T = Temper, A = Age, TO = Type of Offence, SL = Sentence Length

* p≤.05

** p≤.01

## Hierarchical regression analyses

Psychopathy was measured using two dimensions (subscales): Primary Psychopathy and Secondary Psychopathy. Two hierarchical regression analyses were conducted with one of the psychopathy dimensions as the criterion variable. All assumptions of regression analyses (i.e., sample size, normality, outliers, multi-collinearity, and normality, linearity, and homoscedasticity of residuals) were investigated. None of the assumptions was violated. Once again, the focus of the discussion will be on the contributions that were of statistical and practical significance. Both the 1% and 5% levels of significance were considered. When performing hierarchical regression analyses, an effect size of .02 is considered to be small, an effect size of .15 is considered to be medium, and an effect size of .35 is considered to be large [100]. Only findings with medium effect sizes will be discussed.

### Hierarchical regression analyses with primary psychopathy as criterion variable

Table 4 depicts the results of the hierarchical regression analysis with Primary Psychopathy as the criterion variable.

It is evident from Table 4 that the combination of the independent (predictor) variables contributed to 55.9% ($F_{11;127}$ = 14.626; p≤,001) of the variance in the Primary Psychopathy scores of the sample. This finding was statistically significant at the 1% level, and the large corresponding effect size ($f^2$ = 1.070) suggested that it is of considerable practical significance. As a set of predictor variables, the SCSM subscales (Impulsivity, Simple Tasks, Risk-Seeking, Self-Centredness and Temper) were responsible for 31.3% of the variance in the Primary Psychopathy scores of the incarcerated female offenders. This finding was statistically significant at the 1% level, and the corresponding large effect size ($f^2$ = .710) suggested that it is of considerable practical significance. Table 4 further indicates that Impulsivity, Simple Tasks, Risk-Seeking, and Self-Centredness, respectively, explained 15.8% ($F_{7;131}$ = 34.728; p≤.01; $f^2$ = .265), 17.0%

**Table 4. Contributions of age, type of offence, sentence length, MMS subscales, and SCSM subscales to $R^2$ with primary psychopathy as criterion variable.**

| Variables in equation | $R^2$ | Contribution to $R^2$: full minus reduced model | $F$ | $f^2$ |
|---|---|---|---|---|
| 1. [Demographics] + [MMS] + [SCSM] | .559 | 1–7 = .313 | **18.028**\*\* | **.710** |
| 2. [Demographics] + [MMS] + I | .404 | 2–7 = .158 | **34.728**\*\* | **.265** |
| 3. [Demographics] + [MMS] + ST | .416 | 3–7 = .170 | **38.134**\*\* | **.291** |
| 4. [Demographics] + [MMS] + RS | .369 | 4–7 = .123 | **25.536**\*\* | **.195** |
| 5. [Demographics] + [MMS] + SC | .498 | 5–7 = .252 | **65.761**\*\* | **.502** |
| 6. [Demographics] + [MMS] + T | .316 | 6–7 = .070 | 13.406\*\* | .102 |
| 7. [Demographics] + [MMS] | .246 | | | |
| 8. [Demographics] + [SCSM] + [MMS] | .559 | 8–12 = .042 | 4.032\*\* | .095 |
| 9. [Demographics] + [SCSM] + ROC | .554 | 9–12 = .037 | 10.702\*\* | .083 |
| 10. [Demographics] + [SCSM] + D | .552 | 10–12 = .035 | 10.078\*\* | .078 |
| 11. [Demographics] + [SCSM] + C | .542 | 12–12 = .025 | 7.041\*\* | .055 |
| 12. [Demographics] + [SCSM] | .517 | | | |
| 13. [MMS] + [SCSM] + [Demographics] | .559 | 14–17 = .043 | 4.128\*\* | .098 |
| 14. [MMS] + [SCSM] + A | .543 | 15–17 = .027 | 7.621\*\* | .059 |
| 15. [MMS] + [SCSM] + TO | .517 | 16–17 = .001 | .267 | .002 |
| 16. [MMS] + [SCSM] + SL | .540 | 17–17 = .024 | 6.730\* | .052 |
| 17. [MMS] + [SCSM] | .516 | | | |

Key: A = Age, TO = Type of Offence, SL = Sentence Length, ROC = Regulation of Cognition, D = Knowledge of Cognition (Declarative), C = Knowledge of Cognition (Conditional), I = Impulsivity, ST = Simple Tasks, RS = Risk-Seeking, SC = Self-Centredness, T = Temper

\*\*p≤.01

\*p≤.05

($F_{7;131}$ = 38.134; $p$≤.01; $f^2$ = .291), 12.3% ($F_{7;131}$ = 25.536; $p$≤.01; $f^2$ = .195), and 25.2% ($F_{7;131}$ = 65.761; $p$≤.01; $f^2$ = .502) of the variance in the participants' Primary Psychopathy scores. The relevant effect sizes indicated that these findings were of practical significance.

## Hierarchical regression analyses with secondary psychopathy as criterion variable

The results of the hierarchical regression analysis with Secondary Psychopathy as the criterion variable are reported in Table 5.

It is evident from Table 5 that the combination of the independent (predictor) variables contributed to 60.9% ($F_{11;127}$ = 18.014; $p$≤,001) of the variance in the Secondary Psychopathy scores of the sample. This finding was statistically significant at the 1% level, and the large corresponding effect size ($f^2$ = .720) suggested that it is of considerable practical significance. As a set of predictor variables, the SCSM subscales (Impulsivity, Simple Tasks, Risk-Seeking, Self-Centredness and Temper) were responsible for 41.7% of the variance in the Secondary Psychopathy scores of the incarcerated female offenders. This finding was statistically significant at the 1% level, and the corresponding effect size ($f^2$ = 1.066) suggested that it is of considerable practical significance. Table 5 further indicates that Impulsivity, Simple Tasks, Risk-Seeking, Self-Centredness, and Temper, respectively, explained 27.1% ($F_{7;131}$ = 66.110; $p$≤.01; $f^2$ = .505), 23.6% ($F_{7;131}$ = 54.049; $p$≤.01; $f^2$ = .413), 19.5% ($F_{7;131}$ = 41.672; $p$≤.01; $f^2$ = .318), 18.6% ($F_{7;131}$ = 39.174; $p$≤.01; $f^2$ = .299), and 30.9% ($F_{7;131}$ = 81.120; $p$≤.01; $f^2$ = .619) of the variance in the participants' Secondary Psychopathy scores. The relevant effect sizes suggested that these findings were of practical significance.

**Table 5. Contributions of age, type of offence, sentence length, MMS subscales, and SCSM subscales to $R^2$ with secondary psychopathy as criterion variable.**

| Variables in equation | $R^2$ | Contribution to $R^2$: full minus reduced model | $F$ | $f^2$ |
|---|---|---|---|---|
| 1. [Demographics] + [MMS] + [SCSM] | .609 | 1–7 = .417 | **27.089**\** | **1.066** |
| 2. [Demographics] + [MMS] + I | .463 | 2–7 = .271 | **66.110**\** | **.505** |
| 3. [Demographics] + [MMS] + ST | .428 | 3–7 = .236 | **54.049**\** | **.413** |
| 4. [Demographics] + [MMS] + RS | .387 | 4–7 = .195 | **41.672**\** | **.318** |
| 5. [Demographics] + [MMS] + SC | .378 | 5–7 = .186 | **39.174**\** | **.299** |
| 6. [Demographics] + [MMS] + T | .501 | 6–7 = .309 | **81.120**\** | **.619** |
| 7. [Demographics] + [MMS] | .192 | | | |
| 8. [Demographics] + [SCSM] + [MMS] | .609 | 8–12 = .022 | 2.382** | .056 |
| 9. [Demographics] + [SCSM] + ROC | .600 | 9–12 = .013 | 4.193* | .033 |
| 10. [Demographics] + [SCSM] + D | .609 | 10–12 = .022 | 7.258** | .056 |
| 11. [Demographics] + [SCSM] + C | .598 | 12–12 = .011 | 3.530 | .027 |
| 12. [Demographics] + [SCSM] | .587 | | | |
| 13. [MMS] + [SCSM] + [Demographics] | .609 | 14–17 = .012 | -.123 | .031 |
| 14. [MMS] + [SCSM] + A | .598 | 15–17 = .001 | -.025 | .002 |
| 15. [MMS] + [SCSM] + TO | .597 | 16–17 = .000 | - | - |
| 16. [MMS] + [SCSM] + SL | .608 | 17–17 = .011 | -.281 | .028 |
| 17. [MMS] + [SCSM] | .597 | | | |

Key: A = Age, TO = Type of Offence, SL = Sentence Length, ROC = Regulation of Cognition, D = Knowledge of Cognition (Declarative), C = Knowledge of Cognition (Conditional), I = Impulsivity, ST = Simple Tasks, RS = Risk-Seeking, SC = Self-Centredness, T = Temper

\**p≤.01

\*p≤.05

## Discussion

Statistically and practically significant positive correlations were found between the LSRP dimensions (Primary Psychopathy and Secondary Psychopathy) and SCSM subscales (Impulsivity, Simple Tasks, Risk-Seeking, Self-Centredness and Temper). These findings suggest that as the incarcerated female offenders' levels of impulsivity, preference for simple tasks, risk-seeking, self-centredness and temper increase, their primary psychopathy increases. Similarly, these findings indicate that as the incarcerated female offenders' levels of impulsivity, preference for simple tasks, risk-seeking, self-centredness and temper increase, their secondary psychopathy increases. These findings are congruous with previous studies that have found statistically significant correlations between self-control and psychopathy [53, 101, 102], with notable correlations identified between self-control and secondary psychopathy [9]. This is comparable with Prado et al.'s [103] result that those presenting with lower self-control (i.e., behaviour culminating in increased impulsivity, preference for simple tasks, risk-seeking, self-centredness and temper) are more likely to present with secondary psychopathic traits. This seems to suggest that female incarcerated offenders' lower degree of self-control may be more likely explained by the behaviours associated with their secondary psychopathic traits (such as sensation-seeking, disinhibition, impulsivity, lack of responsibility and antisocial lifestyle) rather than their affective and interpersonal characteristics. Furthermore, the present study found three particularly significant correlations between Secondary Psychopathy and Impulsivity, Primary Psychopathy and Self-Centredness, as well as Secondary Psychopathy and Temper.

A study by Fekih-Romdhane et al. [9] supports the finding of a significant positive relationship between Impulsivity and Secondary Psychopathy. This relationship can be explained by

the finding that impulsivity is often considered a cardinal feature, particularly associated with the behavioural facets of psychopathy [56]. Prado et al.'s [103] result further confirmed Snowden and Gray's [104] observation that adult offenders with higher scores on the secondary dimension of psychopathy demonstrate elevated impulsivity, while those with higher primary psychopathy scores exhibit reduced impulsivity, suggesting notable differences between the impulsive nature of the two variants of psychopathy. In practical terms, this may indicate that female incarcerated offenders presenting with secondary psychopathy may engage more frequently in behaviours considered to be spur-of-the-moment or last-minute and may, therefore, not take the time to consider the consequences of their behaviours but instead take immediate action resulting in immediate gratification.

Furthermore, as with the results obtained by Armstrong et al. [53] (i.e., significant correlations between Self-Centredness and Primary Psychopathy and Temper and Secondary Psychopathy), these findings can be attributed to literature whereby primary psychopaths are theorised to be emotionally detached, which is supposed to result from an intrinsic deficit in one's emotional processing and are thus more inclined to be callous, self-centred, and lacking empathy [105, 106]. In practical terms, this may indicate that female incarcerated offenders presenting with primary psychopathy may have poorer interpersonal relationships with others as they may not consider others' perspectives, feelings or intentions and may prioritise their own well-being over others. Moreover, secondary psychopaths are conjectured to develop a proneness to poorly regulated negative affect often characterised by high levels of hostility and aggression and are thus more inclined to quick temper [103, 107]. In practical terms, this may indicate that female incarcerated offenders presenting with secondary psychopathy may behave with increased aggressive behaviours toward others without necessary provocation or may respond in an increasingly aggravated manner to a degree unnecessitated to the situation at hand.

Contrary to previous research, the present study found significant correlations between the Simple Tasks and Risk-Seeking dimensions of the SCSM with secondary psychopathy. These findings may not be wholly unjustifiable as secondary psychopaths are often characterised by impulsive, irresponsible, and antisocial behaviours [106]. They may, therefore, be more inclined to dislike challenging projects and take pleasure in risk-seeking activities. It could be suggested that the correlations between Simple Tasks and Risk-Seeking in relation to psychopathy may be specific associations unique to the South African female offender population.

Statistically and practically significant negative correlations were also identified between the LSRP dimensions (Primary Psychopathy and Secondary Psychopathy) and Age. These findings suggest that as the incarcerated female offenders' age increases, psychopathy seems to decrease. These findings are congruous with previous studies that have found negative correlations between age and psychopathy [64, 65, 67, 108, 109]. According to Hartung et al. [66], socially adverse personality characteristics, for example, psychopathy, uniformly decrease with increasing age. Notably, the present study determined Age significantly correlated with Primary Psychopathy; the same was found to be true in a recent study among female offenders [109]. In practical terms, this may indicate that if female incarcerated offenders present with increased behaviours associated with primary psychopathic traits (such as callousness, grandiosity, fearlessness, lack of remorse and guilt), the younger they may be. Thus, female offenders presenting with primary psychopathy may be less likely to exhibit callous, grandiose, and remorseless behaviours as they grow older.

The hierarchical regression analyses revealed that the combination of the predictor variables (Age, Type of Offence, Sentence Length, MMS subscales, and SCSM subscales) statistically and practically significantly predicted both dimensions of the LSRP (i.e., Primary Psychopathy and Secondary Psychopathy). These findings are congruous with the literature.

In a study conducted by Jonason and Tost [102], self-control was reported to be a robust regression coefficient of psychopathy, with a partial variance associated with the participant's age [110]. Several studies have also found low self-control to consistently be a significant predictor of psychopathy among diverse populations [53, 101, 111–115].

Significant predictors of primary psychopathy were self-centredness (25.2%), simple tasks (17%), impulsivity (15.8%), and risk-seeking (12.3%). In practical terms, these findings suggest that as female incarcerated offenders' egocentricity, preference for simple tasks, impulsivity and recklessness increase, their affective and interpersonal psychopathic traits increase. For secondary psychopathy, temper (30.9%), impulsivity (27.1%), simple tasks (23.6%), risk-seeking (19.5%), and self-centredness (18.6%) were significant predictors. In practical terms, these findings suggest that as female incarcerated offenders' temper, impulsivity, preference for simple tasks, recklessness and egocentricity increase, their affective and interpersonal psychopathic traits increase. These findings are congruous with researchers' postulations that impulsivity is a significant contributor to individuals' psychopathic traits and antisocial behaviours [9, 109, 116, 117]. These findings are further supported by literature, where impulsivity has been found to be associated with the socially deviant features of psychopathy in offenders [118], more specifically with poor impulse control of one's executive function measures [119, 120], thrill and experience seeking traits, and disinhibition [121], characteristics typically ascribed to the secondary dimension of psychopathy. Furthermore, researchers have utilised impulsivity (i.e., a lack of self-control and cognitive complexity) to explain violence in female offenders [109]. This supports prior research that has shown violent offenders display deficits within their executive functioning skills, such as inhibitory control and cognitive flexibility [122], characteristics typically attributed to the primary dimension of psychopathy.

The findings of several significant individual predictors of psychopathy (e.g., simple tasks, risk-seeking, self-centredness and temper) appear to be a novel result of this study, as there is currently no literature available to support the significant variance between these individual predictor variables and psychopathy (EBSCOHost, 2022 August). This seems to suggest that previous research has likely not explored individual factors of self-control as possible predictors of psychopathy. Therefore, additional research is needed to duplicate the findings of simple tasks, risk-seeking, and self-centredness as significant predictor variables amongst female incarcerated offenders.

## Limitations of the study

Several studies have detailed the psychopathy of incarcerated offenders [16]. However, a search on EBSCOHost (2022, August) did not deliver results on any previous or similar studies that investigated the predictors of psychopathy amongst female incarcerated offenders, specifically within the South African context. This was a pertinent limitation of the study, considering that there was no previous literature regarding this population group to draw upon or compare the results.

The generalisability of the results to the broader female offender population in South Africa was another significant limitation of this study. The paradigm of quantitative research places emphasis on the generalisation of results. With the use of convenience sampling, the sample was not representative of the population, and therefore, these results are not generalisable [80] to other populations of offenders.

## Recommendations for future research

There are several important recommendations regarding potential future research, particularly regarding incarcerated female offenders. Firstly, South African researchers must undertake to

conduct sustained research not only amongst incarcerated offender populations but, more specifically, female incarcerated offenders across correctional centres in South Africa. The Department of Correctional Services Research Agenda (2019–2023) [123, p. 3] unequivocally states that:

> Research in corrections has a high value to society. It has provided important information about incarceration trends for planning and identifying risk factors to improve security in corrections. Research has led to significant discoveries, the development of new ways of rehabilitating offenders and improvements in correctional care.

Without continued research, it would be difficult to support offenders during their rehabilitation within the correctional environment over and above the broader community when released [76]. Although international and national interest in correctional-based research is rapidly expanding [124], research on incarcerated female offenders continues to lag. Thus, for this study, a paucity of research continues to exist amongst South African female incarcerated offenders regarding psychopathy and cannot be drawn upon.

This study investigated the predictors of female offenders' psychopathy within the BMCC in Kroonstad, Free State, and the JCC in Johannesburg, Gauteng alone. This resulted in a limited sample size which was also not representative of the general South African female offender population. To adequately gauge how these predictors of psychopathy (impulsivity, simple tasks, risk-seeking, self-centredness and temper) drawn from international studies apply to female offenders in South Africa, a broader investigation of inquiry, including a larger sample obtained from several correctional centres, could yield more findings regarding this topic.

Also, the use of probability sampling is recommended for future research endeavours, as it would yield a representative sample of the larger population of South African female offenders that would be beneficial in generalising future results to the broader correctional context [80]. This would result in a better understanding of how the predictors of psychopathy interact with this concept in the broader spectrum of the multicultural South African landscape.

Furthermore, future research may investigate other possible predictors of psychopathy among female offenders that were not necessarily analysed as they were beyond the scope of the current study. This includes family history, aggressive behaviour towards others, previous convictions (i.e., first-time or repeat offenders), gang involvement, personality disorders and substance abuse [8, 10, 35, 62, 125–128]. This may advance our understanding of the best predictors of psychopathy amongst incarcerated female offenders in the South African context.

Additionally, researchers have found support for different factor solutions of the LSRP across samples [16]. Therefore, future researchers can conduct similar research with a larger and more representative sample to validate the applicability of a different factor solution measure of female offenders' psychopathy within the South African context.

## Conclusion

The present study has identified correlations between several predictor and criterion variables. Statistically and practically significant correlations were identified between Primary Psychopathy and Impulsivity, Simple Tasks, Risk-Seeking and Self-Centredness, and between Secondary Psychopathy and Impulsivity, Simple Tasks, Risk-Seeking, Self-Centredness and Temper. The correlations between the other predictor variables, for example, Age, Sentence Length, and the MMS subscales with any of the criterion variables did not reach statistical and/or practical significance. Regarding the regression analyses, the percentage of the variance these predictor variables explained was statistically significant but of limited practical significance. Therefore,

this study found that the individual predictor variables or the combination(s) of predictor variables, specifically the dimensions of self-control, statistically and practically significantly contributed to the variance of psychopathy amongst incarcerated female offenders in the South African context. This was the goal of this research study. Limitations and recommendations for future research have been noted. Regardless of the limitations, the study provided information contributing to the existing literature on incarcerated female offenders.

## Supporting information

**S1 Data. Minimal dataset.**
(SAV)

## Acknowledgments

We would like to thank all volunteers for their participation in this study. We further express our gratitude to the Department of Correctional Services for assistance with the recruitment and assessment of study participants.

## Author Contributions

**Conceptualization:** Judite Danielle de Oliveira.

**Data curation:** Judite Danielle de Oliveira.

**Formal analysis:** Judite Danielle de Oliveira, Jacques Jordaan.

**Supervision:** Jacques Jordaan, Matthew Cronjé.

**Writing – original draft:** Judite Danielle de Oliveira.

**Writing – review & editing:** Judite Danielle de Oliveira, Jacques Jordaan, Matthew Cronjé.

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
