## [Decision Letter · Decision Letter 0]

9 Nov 2023

PONE-D-23-20668Morality, self-control, age, type of offence and sentence length as predictors of psychopathy amongst female incarcerated offenders in South AfricaPLOS ONE

Dear Dr. Jordaan,

Thank you for submitting your manuscript to PLOS ONE. After careful consideration, we feel that it has merit but does not fully meet PLOS ONE’s publication criteria as it currently stands. Therefore, we invite you to submit a revised version of the manuscript that addresses the points raised during the review process.

We look forward to receiving your revised manuscript.

Kind regards,

Vincenzo De Luca

Academic Editor

PLOS ONE

Journal Requirements:

2. Please provide additional information regarding the considerations  made for the prisoners included in this study. For instance, please discuss whether participants were able to opt out of the study and whether individuals who did not participate receive the same treatment offered to participants.

Reviewers' comments:

Reviewer's Responses to Questions

**Comments to the Author**

1. Is the manuscript technically sound, and do the data support the conclusions?

Reviewer #1: Yes

2. Has the statistical analysis been performed appropriately and rigorously? 

Reviewer #1: Yes

3. Have the authors made all data underlying the findings in their manuscript fully available?

Reviewer #1: Yes

4. Is the manuscript presented in an intelligible fashion and written in standard English?

Reviewer #1: Yes

5. Review Comments to the Author

Reviewer #1: The study by Oliveira et al, examined psychopathic traits of female incarcerated offenders in South Africa. Using a quantitative and correlational approach, 139 female from South African correctional centers were analyzed. The research uncovers specific predictor variables, including impulsivity, simple tasks, risk-seeking behavior, and self-centeredness, significantly explaining both primary and secondary psychopathy. The findings are intriguing. However, I recommend a comprehensive restructuring of the paper before publication. A reorganization could enhance the clarity, coherence, and impact of the research, ensuring a more robust presentation of the valuable insights gained from the study.

Overall, the introduction is excessively long, particularly regarding the decrease in the offender population. With 10 pages dedicated to the introduction, it becomes challenging to grasp the study's significance, objectives, and scientific advancements. Simplifying this section is crucial to enhance clarity and make the content more accessible to readers.

The following phrase is quite similar to the original text. I recommend a more direct and concise approach to rewriting the paragraph.

“According to Levenson [17], psychopathy should not be viewed as a disorder characterised by a deficiency in neurological systems mediated by one’s own relative anxiety or harm avoidance but should be conceptualised as a pattern of intrinsically antisocial behaviour that is based on judgements relating to the relative importance of one’s own wishes and the rights and well-being of others. Levenson”.

In the methods, the authors mention that “The questionnaires took one to one and a half hours to complete depending on the literacy level of the offenders”. Could you clarify if any evaluator instructed individuals to read the questionnaire or assisted them in this process?

The authors mentioned that “The LSRP focuses on two distinct dimensions of psychopathy, namely primary psychopathy and secondary psychopathy:” What does the definition of primary and secondary psychopathology mean from a psychiatric perspective? It is difficult to translate this concept into the domain of modern psychiatry, especially with diagnoses primarily conducted through DSM-V. Could you include an explanation of why you employed this definition? Could you also explain if primary and secondary psychopathology are related to mental health diagnoses such as Major Depressive Disorder or Anxiety Disorders?

The results section is overly extensive. It is difficult to read and to follow the main point of the study. I recommend a complete rewrite for conciseness. Additionally, consider using subtitles related to the findings, not just the authors' analysis.

Tables 4 and 5 are challenging to follow. I recommend presenting these results in a different format for better clarity.

6. PLOS authors have the option to publish the peer review history of their article (what does this mean?). If published, this will include your full peer review and any attached files.

Reviewer #1: **Yes: **Randriely Merscher Sobreira de Lima

---

## [Author Response · Author response to Decision Letter 0]

24 Nov 2023

Dear Editors and Reviewers

Thank you for reviewing our manuscript. We have uploaded our responses and feedback in a feedback table. We hope that you find the changes and feedback in the feedback table satisfactory.

Kind regards.

---

## [Decision Letter · Decision Letter 1]

10 Jan 2024

PONE-D-23-20668R1Morality, self-control, age, type of offence and sentence length as predictors of psychopathy amongst female incarcerated offenders in South AfricaPLOS ONE

Dear Dr. Jordaan,

Thank you for submitting your manuscript to PLOS ONE. After careful consideration, we feel that it has merit but does not fully meet PLOS ONE’s publication criteria as it currently stands. Therefore, we invite you to submit a revised version of the manuscript that addresses the points raised during the review process.

We look forward to receiving your revised manuscript.

Kind regards,

Vincenzo De Luca

Academic Editor

PLOS ONE

Reviewers' comments:

Reviewer's Responses to Questions

**Comments to the Author**

1. If the authors have adequately addressed your comments raised in a previous round of review and you feel that this manuscript is now acceptable for publication, you may indicate that here to bypass the “Comments to the Author” section, enter your conflict of interest statement in the “Confidential to Editor” section, and submit your "Accept" recommendation.

Reviewer #1: All comments have been addressed

2. Is the manuscript technically sound, and do the data support the conclusions?

Reviewer #1: Yes

3. Has the statistical analysis been performed appropriately and rigorously? 

Reviewer #1: Yes

4. Have the authors made all data underlying the findings in their manuscript fully available?

Reviewer #1: Yes

5. Is the manuscript presented in an intelligible fashion and written in standard English?

Reviewer #1: Yes

6. Review Comments to the Author

Reviewer #1: Oliveira and colleagues made a substantial improvement in the study. However, there are still some aspects that need further refinement.

- The introduction is too long with five pages. Try to make it shorter and more to the point, presenting the main ideas clearly and concisely.

- I suggest incorporating the Purpose of the study and the Research aim and questions in the introduction.

- For Tables 4 and 5, I recommend replacing the summarizing variables (i.e., “[A + TO + SL]”) with what they represent. For example, [A + TO + SL] = Demographics, [ROC + D + C] = MMS, and [I + ST + RS + SC + T] = SCSM. This way, it’s clear to the reader why those variables are grouped the way they are.

7. PLOS authors have the option to publish the peer review history of their article (what does this mean?). If published, this will include your full peer review and any attached files.

Reviewer #1: **Yes: **Randriely Merscher Sobreira de Lima

---

## [Author Response · Author response to Decision Letter 1]

24 Jan 2024

Dear Reviewers

Thank you for taking the time to review our manuscript. We made the necessary changes and have provided feedback in a feedback table that has been uploaded.

Kind regards.

---

## [Decision Letter · Decision Letter 2]

19 Feb 2024

Morality, self-control, age, type of offence and sentence length as predictors of psychopathy amongst female incarcerated offenders in South Africa

PONE-D-23-20668R2

Dear Dr. Jordaan,

We’re pleased to inform you that your manuscript has been judged scientifically suitable for publication and will be formally accepted for publication once it meets all outstanding technical requirements.

Kind regards,

Vincenzo De Luca

Academic Editor

PLOS ONE

Additional Editor Comments (optional):

Reviewers' comments:

Reviewer's Responses to Questions

**Comments to the Author**

1. If the authors have adequately addressed your comments raised in a previous round of review and you feel that this manuscript is now acceptable for publication, you may indicate that here to bypass the “Comments to the Author” section, enter your conflict of interest statement in the “Confidential to Editor” section, and submit your "Accept" recommendation.

Reviewer #1: All comments have been addressed

2. Is the manuscript technically sound, and do the data support the conclusions?

Reviewer #1: Yes

3. Has the statistical analysis been performed appropriately and rigorously? 

Reviewer #1: Yes

4. Have the authors made all data underlying the findings in their manuscript fully available?

Reviewer #1: Yes

5. Is the manuscript presented in an intelligible fashion and written in standard English?

Reviewer #1: Yes

6. Review Comments to the Author

Reviewer #1: Oliveira and colleagues have made substantial improvements to their study, addressing all of my previous comments. I believe the manuscript is now ready for publication.

7. PLOS authors have the option to publish the peer review history of their article (what does this mean?). If published, this will include your full peer review and any attached files.

Reviewer #1: **Yes: **Randriely Merscher Sobreira de Lima

---

## [Editor Report · Acceptance letter]

14 Mar 2024

PONE-D-23-20668R2 

PLOS ONE

Dear Dr. Jordaan, 

I'm pleased to inform you that your manuscript has been deemed suitable for publication in PLOS ONE. Congratulations! Your manuscript is now being handed over to our production team.

Kind regards, 

on behalf of

Dr. Vincenzo De Luca 

Academic Editor

PLOS ONE